# A Systematic Review of Deep-Learning Methods for Intracranial Aneurysm Detection in CT Angiography

**DOI:** 10.3390/biomedicines11112921

**Published:** 2023-10-28

**Authors:** Žiga Bizjak, Žiga Špiclin

**Affiliations:** Laboratory of Imaging Technologies, Faculty of Electrical Engineering, University of Ljubljana, 1000 Ljubljana, Slovenia

**Keywords:** computed tomography angiography, CTA, aneurysm detection, meta-analysis, PRISMA, QUADAS-2, sensitivity, specificity, false positives per image, healthy controls, evaluation guidelines

## Abstract

**Background:** Subarachnoid hemorrhage resulting from cerebral aneurysm rupture is a significant cause of morbidity and mortality. Early identification of aneurysms on Computed Tomography Angiography (CTA), a frequently used modality for this purpose, is crucial, and artificial intelligence (AI)-based algorithms can improve the detection rate and minimize the intra- and inter-rater variability. Thus, a systematic review and meta-analysis were conducted to assess the diagnostic accuracy of deep-learning-based AI algorithms in detecting cerebral aneurysms using CTA. **Methods:** PubMed (MEDLINE), Embase, and the Cochrane Library were searched from January 2015 to July 2023. Eligibility criteria involved studies using fully automated and semi-automatic deep-learning algorithms for detecting cerebral aneurysms on the CTA modality. Eligible studies were assessed using the Preferred Reporting Items for Systematic Reviews and Meta-Analysis (PRISMA) guidelines and the Quality Assessment of Diagnostic Accuracy Studies 2 (QUADAS-2) tool. A diagnostic accuracy meta-analysis was conducted to estimate pooled lesion-level sensitivity, size-dependent lesion-level sensitivity, patient-level specificity, and the number of false positives per image. An enhanced FROC curve was utilized to facilitate comparisons between the studies. **Results:** Fifteen eligible studies were assessed. The findings indicated that the methods exhibited high pooled sensitivity (0.87, 95% confidence interval: 0.835 to 0.91) in detecting intracranial aneurysms at the lesion level. Patient-level sensitivity was not reported due to the lack of a unified patient-level sensitivity definition. Only five studies involved a control group (healthy subjects), whereas two provided information on detection specificity. Moreover, the analysis of size-dependent sensitivity reported in eight studies revealed that the average sensitivity for small aneurysms (<3 mm) was rather low (0.56). **Conclusions:** The studies included in the analysis exhibited a high level of accuracy in detecting intracranial aneurysms larger than 3 mm in size. Nonetheless, there is a notable gap that necessitates increased attention and research focus on the detection of smaller aneurysms, the use of a common test dataset, and an evaluation of a consistent set of performance metrics.

## 1. Introduction

Most IAs are small, and it is estimated that 50–80% do not rupture during a person’s lifetime. Nevertheless, rupture of IA is one of the most common causes of subarachnoid hemorrhage (SAH) [1], a condition with approximately 50% mortality rate [2]. The global annual risk of IA rupture is estimated at 0.95–2.0%, while the risk of complications during or due to treatment is estimated at 5.3% and 6.3% for endovascular coiling and neurosurgical clipping, respectively. The decision on how to treat SAH is determined by considering treatment risk and other relevant factors [3], concurrently with the emergence of new medicines offering promising avenues for improved SAH treatment [4]. The risk of rupture for newly discovered small IAs with a diameter up to 5 mm is less than 1%, and progressively increases with age and possible IA growth, whereas for newly discovered larger IAs, the rupture risk is much higher. Early detection of IAs is required to open a window of opportunity to mitigate rupture risk by determining the optimal time and type of treatment.

The screening for IAs in clinical practice is performed mostly by CTA and MRA imaging. The IAs are detected via visual assessment of the scans, which is a cumbersome task subject to human error. Even skilled radiologists achieve a rather low sensitivity for small IAs, for instance, from 64 to 74.1% for CTAs (IA diameter ≤ 3 mm) [5] and from 70 to 92.8% (IA diameter ≤ 5 mm) [6], which seems could be achieved or even improved using computer-assisted deep learning and artificial intelligence-based approaches [7]. For instance, in a recent study, Yang et al. [8] used such a computer-aided aneurysm detection tool and found that 8 out of 649 aneurysms (1.2%) had been overlooked in the initial radiologic reports. Recent research also suggests that using deep-learning models (DLMs) to assist radiologists in detecting secondary intracranial aneurysms can improve their detection sensitivity [9].

The computer-assisted IA detection system also seems to be a solution to the relative shortage of experienced radiologists compared with the increasing demand for imaging studies [10]. The earliest automated computerized scheme of IA detection was reported in 2004, which achieved high accuracy in a small sample [11]. Recently, on account of progress in deep-learning technology, the volume of research on DLMs for detecting IAs has increased substantially. According to a meta-analysis [12], the diagnostic performance of DLMs was demonstrated to be equivalent to that of healthcare professionals in classifying diseases using medical imaging.

However, even though DLMs were proven to be useful, there are only a few studies on automatic detection of IAs from CTA scans, despite its widespread use, which is likely due to the severity of the problem. Among the reasons is that CTA depicts poor case-to-case and spatially varying vessel-to-background contrast, bony structures with similar intensity as the vessels, and vessel-like streak artifacts, to name a few. As a result, most of the existing research has concentrated on detecting IA aneurysms using MRA and DSA modalities. In a recent review paper [7], the authors included 19 studies involving MRA, 11 studies involving DSA, and merely 4 studies that incorporated CTA as the imaging modality.

Recent two review studies [7,13] focused on all three commonly used modalities (MRA, CTA, and DSA) for IA detection, but have not provided a comprehensive review and analysis of methods utilizing the CTA modality in corresponding test datasets. Specifically, Din et al. [7] reviewed 3, and Gu et al. [13] reviewed 6 research papers, while our review of the field identified 15 eligible research papers, thus indicating recent rapid progress in the field since the aforementioned two studies were published. To enhance the understanding of state of the art in IA detection using CTA, we have therefore performed a systematic review and meta-analysis, using established methodologies. We specifically focus on the structure and size of test datasets employed in the reviewed studies, as well as the consistency and validity of the reported evaluation metrics. By examining a larger pool of studies, our analysis aims to elucidate the current state of developments in the field, offering crucial guidance for shaping future research endeavors and, ultimately, influencing evidence-based clinical practices.

## 2. Materials and Methods

This review was performed in adherence with the guidelines of Preferred Reporting Items for Systematic Reviews and Meta-Analysis (PRISMA) [14], with insights from Cochrane review methodology for establishing study inclusion criteria [15], conducting the study search, and assessing study quality [16].

### 2.1. Literature Search

A comprehensive yet sensitive search was conducted, utilizing subject headings with exploded terms and without imposing any language restrictions [14]. The search terms were applied across various databases, including Embase, MEDLINE, Web of Science, and the Cochrane Register, aiming to extract original research articles (the detailed search strategy is available in Appendix A). In addition to the database search, the bibliography of all relevant authors was meticulously screened to identify any potentially missed articles. To ensure the reliability and validity of the findings, non-peer-reviewed journal articles were excluded from the analysis. Furthermore, the introduction of deep-learning methods to medical imaging applications has progressively increased since 2015 [17]. Therefore, studies published from January 2015 to July 2023 were included in the review process.

### 2.2. Selection Criteria

Our study employed specific inclusion and exclusion criteria to ensure the selection of relevant research. Inclusion criteria were established as follows: (i) the study had to include patients with confirmed one or more IAs through standard-of-care diagnosis or expert consensus; (ii) the study utilized a DLM for the detection of IAs; (iii) the study had to incorporate the CTA modality in their test image dataset.

Conversely, certain exclusion criteria were applied to eliminate studies that did not meet our research objectives. These criteria included studies written in languages other than English, studies that did not involve human subjects, pilot studies, conference papers, abstract-only publications, and letters.

Finally, an eligibility assessment was performed to guarantee the inclusion of appropriate studies. For this purpose, two researchers conducted independent reviews of the titles and abstracts of the selected studies. In instances of disagreements, a senior researcher served as an arbitrator, making the final decision. Additionally, when the same deep-learning model was evaluated in different studies and on different test datasets, each publication was treated as a distinct research instance and included in our study. This approach ensured a comprehensive coverage of relevant and up-to-date findings in our analysis.

### 2.3. Data Extraction and Quality Assessment

Data extraction for analyses involved two individual researchers and subsequent verification by a senior researcher. The collected data encompassed the following variables: publication year, data source (multi or single center), modality used (whether authors employed other modalities besides CTA), study design (retrospective or prospective), used deep-learning model, quantity of CTA images, number of aneurysms in the training and test sets, average size of the aneurysms or any reported related measurement, performance based on aneurysm size if reported by the authors, inclusion of healthy subjects in the test set, and performance by collecting evaluation metric values, such as patient-level sensitivity and lesion-level sensitivity, patient-level specificity, and number of false positives per image.

The evaluation of study quality involved the implementation of the QUADAS-2 (Quality Assessment of Diagnostic Accuracy Studies) tool, a widely acknowledged framework for assessing methodological rigor and bias in diagnostic accuracy studies [18]. Two proficient researchers conducted the assessment independently, meticulously examining aspects like patient selection, index tests, reference standards, and study flow and timing. In cases of disagreements, the senior author acted as an arbitrator, contributing to a comprehensive and objective quality assessment process.

### 2.4. Data Synthesis and Statistical Analysis

The main emphasis of this review study was on the primary outcome measures, which centered around the diagnostic test accuracy metrics of deep-learning methods for detecting IAs on CTA scans. To analyze the data effectively, we employed two units of analysis: (i) patient-level and (ii) lesion-level analysis. Using the published study data, we constructed 2 × 2 confusion matrices for hold-out test datasets. From these matrices, we calculated the primary diagnostic accuracy measures of patient- and lesion-level sensitivity (recall) and patient-level specificity.

Patient-level sensitivity was defined as the number of true positive (TP) scans, where all IAs were correctly identified, divided by the total number of scans, including IAs. Lesion-level sensitivity, on the other hand, was defined as the number of true positive IAs divided by the total number of IAs.

In studies where both internal and external test performance measures were reported, we prioritized the use of external test data to assess performance accuracy. These external test datasets had to be obtained from a distinct institution, typically located in a different geographic region compared to where the training datasets were sourced. Conversely, internal test datasets were considered those obtained from the same institution as the one providing the training datasets.

Deriving specificity required per-patient data, as the number of true negatives cannot be determined on a per-lesion basis. Additionally, we extracted the number of false positives per image (FPs/image), indicating the instances of incorrectly identified IAs within each image.

To facilitate our meta-analysis, we initially assessed the eligible studies based on lesion-level sensitivity and the number of FPs/image, which are among the most commonly reported performance metrics. Additionally, we endeavored to analyze the papers based on patient-level sensitivity and specificity. Considering the observed varying sensitivities in detecting smaller IAs (size < 3 mm), we thoroughly examined all papers that reported lesion-level sensitivity in relation to the size of the IAs.

## 3. Results

### 3.1. Characteristics of Included Studies

Figure 1 provides an overview of the study’s overall flow. Initially, 608 studies that met the search criteria were identified, and from these, 163 full-text articles were assessed for potential eligibility. Eventually, 15 studies published between June 2019 and April 2023 were included in the analysis [8,9,19,20,21,22,23,24,25,26,27,28,29,30,31]. The collective training and testing data encompassed 11,210 subjects, which featured a total of 13,086 IAs. Of these, 9012 cases were utilized for training, while 4074 cases were allocated for validation purposes, both internally (3039 cases) and externally (765 cases). For comprehensive insights, Table 1 and Table 2 present detailed characteristics of the studies, divided into two groups: (i) automatic methods (13/15, 87%) [8,9,19,20,21,22,23,24,25,26,27,28,31], and (ii) semi-automatic methods (2/15, 13%) [29,30]. All studies were retrospective in nature. Among the 15 studies, 13 (13/15, 87%) exclusively employed CTA as the imaging modality. The study by Timmins et al. [29] used MRA for training and CTA for separate evaluation, while another study by Shi et al. [26] utilized DSA and CTA as training modalities. Of the included studies, 7 (7/15, 47%) datasets were acquired from multiple centers, while 8 (8/15, 53%) were obtained from a single center. Out of the 15 studies, 5 (5/15, 33%) included healthy subjects; however, only 2 studies reported patient-level specificity. Notably, only the study by Bo et al. [24] made their data publicly available, whereas, in four studies (4/15, 27%), the authors mentioned the possibility of data sharing upon request; however, in four, to date, none have responded to our email request sent to the corresponding author of each study. Moreover, eight studies (8/15, 53%) reported their results with respect to aneurysm size, while two studies (2/15, 13%) presented results with respect to IA volume.

In the examined studies, 5 (5/15, 33%) studies made use of either the original ResNet model or a modified variant thereof [32,33], and 3 (3/15, 20%) adopted the Deep Medic model [34]. Meanwhile, various other models, such as 3D-UNet, GLIA-NET, 3D DLN-OR, and HeadXNet were each employed only once. Notably, two authors did not specify the deep-learning model they used.

### 3.2. Test Dataset Characteristics

According to Table 2, six studies (6/15, 40%) used an external test dataset (scanner/site not seen during training). Twelve studies (12/15, 80%) used an internal hold-out test dataset. Cross-validation (CV) was not performed in any of the studies. One study [31] did not explicitly describe its test dataset and evaluation protocol.

### 3.3. Sensitivity Analysis

Three studies reported patient-level sensitivity; however, an explicit definition of patient-level sensitivity was not given. To avoid the interpretation of inconsistent metric values, we do not report patient-level sensitivity.

The forest plot depicted in Figure 2 displays the lesion-level sensitivity results for 14 of 15 studies (93%), whereas the study by Meng et al. [31] did not report the corresponding value. The lesion-level sensitivities are reported based on the performance of the test dataset; if there were multiple test datasets in a study, we reported the lesion-level sensitivity on an external dataset. Notably, the lesion-level sensitivity values varied considerably across the studies, ranging from 0.483 to 0.975. Please note that these values were pooled across IAs of all sizes included in each study’s test dataset and may be over-optimistic regarding the detection of small IAs.

Among the included studies, twelve (12/15, 80%) provided information on the average size of IAs, which varied in range from 3.6 to 7.9 mm. Interestingly, two studies reported the volume of IAs but did not include data on IA size. The IA size statistic is generally not sufficient to judge the lesion-level sensitivity of small IAs. Hence, we performed a lesion-level specificity analysis with respect to IA size.

Eight (8/15, 53%) studies reported sufficient data to reliably determine lesion-level sensitivity based on the size of IA [8,19,20,21,23,24,27,29]. It is important to note that while all authors defined small IAs as those with a diameter less than 3 mm, the definitions for medium and large IAs varied across the studies. For a clearer understanding, a graphical presentation illustrating the IA size categories used for IA size groups and the corresponding achieved lesion-level sensitivities and associated IA counts can be found in Figure 3.

The lesion-level sensitivity for the small IAs (<3 mm) varied substantially, i.e., from zero to 0.926. The lesion-level sensitivity is generally higher for the IAs larger than 3 mm in diameter. Most notably, the counts of small IAs used in the studies are rather small, and the small IAs generally represent the minority class with respect to the medium and large IA classes. Hence, the size-pooled lesion-level sensitivities may indeed be biased and over-optimistic for the small aneurysms.

The most important observation is that all studies (except Wang et al. [19]) reported size-based lesion-level sensitivity on internal rather than on the external validation dataset. Therefore, the observed lesion-level sensitivity may be biased and over-optimistic, especially for the small IAs. This is supported by the results of Wang et al. [19], who achieved a rather high overall lesion-level sensitivity of 0.94 (cf. Figure 2), but a corresponding zero value for small IAs.

### 3.4. Specificity Analysis

In 13 studies that reported the number of FPs/image the range spanned from 0.16 to 13.8 (Table 2). On the other hand, the patient-level specificity was reported only in 2/15 (13%) studies that included healthy subject controls. Specifically, Shi et al. [26] achieved a specificity of 0.71, while Park et al. [28] attained quite a low specificity of 0.06. These figures indicate rather poor performance of the respective methods in accurately identifying true negatives and, therefore, necessitate the inclusion of healthy subject control scans and the evaluation of patient-level specificity in future detection method evaluation protocols.

### 3.5. Enhanced FROC Curve

The standard approach for reporting the performance of DLMs in detecting IAs is using the Free-Response Receiver Operating Characteristic (FROC) curve, which plots lesion-level sensitivity with respect to the average number of FPs per image.

For a more general comparison of multiple studies, we enhanced the FROC curve by incorporating two additional pieces of information: (i) the number of IAs in the test dataset (encoded as color) and (ii) the average size of the IAs (encoded as the size of the data point). For studies that reported the IA volume [9,25], we performed an approximate conversion from IA volume to IA size, relying on the conversion introduced by Shahzad et al. [25].

The enhanced FROC curve is depicted in Figure 4. In an ideal scenario, a perfect model would exhibit 0 FPs/image, a lesion-level sensitivity of 1.0, and be represented by a small (=low average IA size) and bright yellow (=high CTA count) data point. The FROC shows that studies by Wang et al. [19] and Wu et al. [22] achieved a good balance between lesion-level sensitivity and FPs/image. The FROC also shows that the higher the lesion-level sensitivity, the more FPs/image.

## 4. Discussion

### 4.1. Summary of Findings

Numerous studies in recent years have applied DLMs for the detection of IAs on CTA images. Despite the inherent challenges associated with IA detection in CTA, such as the presence of bone structures and image artifacts, researchers have continuously reported high lesion-level sensitivity and progressively lower number of false positives per image. However, the diagnostic accuracy of these studies is somewhat limited due to a high risk of bias and concerns regarding the data used. For instance, approximately half of the studies included in this review did not report lesion-level sensitivity with respect to the IA size, and those that did exhibit important limitations such as limited count of small IAs in the test dataset [19,20,29], a small test dataset [19,20,28,30], internal instead of external dataset (all except [19]), poor sensitivity to small IAs [19,20,23,29], or a high number of false positives per image [8,24,27].

Despite the potential of DLMs, many studies lack a comprehensive reporting of evaluation metrics. To enable meaningful comparisons among future research endeavors, it is imperative to establish standardized reporting guidelines for IA detection studies. With this aim in mind, we present the essential performance metrics that should be reported in a comprehensive study of IA detection in later Section 4.6. By adhering to these guidelines, researchers can ensure consistency and facilitate a more objective evaluation and comparison of IA detection methodologies.

### 4.2. Limitations of Reviewed Studies

All studies included in the analysis utilized a retrospective and case-control design. However, it is important to note that the eligibility criteria employed varied across these studies. Some studies excluded IAs smaller than 3 mm, while others excluded patients with specific types or location of IAs. Additionally, it is worth mentioning that not all the methods employed in these studies were fully automatic.

Furthermore, a significant proportion of the studies (10/15, 67%) did not include a control group (healthy subjects) in their analysis to perform patient-level specificity analysis. Among the five studies that did include a control group, only two reported patient-level specificity [26,28]. This lack of reporting and inclusion of healthy subjects (i.e., CTA brain scans without IAs) can introduce spectrum bias and potentially limit the generalizability of the results to real clinical environments. As the results of the two studies show, the attained patient-level specificity was rather low, thus indicating a high chance of false findings using the DLMs.

Although multiple different deep-learning models were tested across studies, to the best of our knowledge, there is no comparative study focused on the variability of the results with respect to the model architecture. The most commonly used model architecture was ResNet or its modifications (5/15 studies), while DeepMedic was used in three studies.

An attempt was made to extract and report patient-level sensitivity. However, among the included studies, only three incorporated information regarding patient-level sensitivity. Regrettably, none of these studies provided a clear and explicit definition of patient-level sensitivity. Consequently, due to the lack of comprehensive and standardized data in this regard, it was not possible to report on patient-level sensitivity within this review.

This and other inconsistencies regarding evaluation metrics were noticed. For instance, most papers accurately reported lesion-level specificity and the number of FPs/image. However, there were inconsistencies in reporting patient-level sensitivity and specificity. To address this issue, we have proposed evaluation metrics and dataset characteristics that should be included in future studies to ensure consistent and comparable reporting of IA detection performance (see Section 4.6).

One significant limitation of all the reviewed studies is that each study validated their DLM exclusively on their private datasets. This is mainly because, until recently, there were no publicly available CTA datasets of IAs. Since the authors may not release their code and datasets for various reasons, for instance, due to patient privacy policies and other regulatory limitations, it is, therefore, impossible to independently verify the effectiveness of their methods and assess the variability of results with respect to the employed deep-learning model architecture. There is one notable exception, namely Bo et al. [24], who publicly released their dataset, complete with a train-test split and relevant external dataset. However, in the present version of the dataset, the IA size information was not given. Thus, size-based lesion-level sensitivity cannot be performed. Nevertheless, this dataset represents a crucial step forward in the objective evaluation of IA detection methods on CTA scans. We highly recommended that all authors employ the dataset provided by Bo et al. [24] (and any other future public datasets) as an external validation dataset to facilitate comparisons between different IA detection methods. By utilizing this dataset and future public datasets, researchers can ensure an objective, reliable, and comprehensive evaluation of their approaches.

### 4.3. Bias and Applicability Assessment

An analysis of the risk of bias and concerns regarding applicability was performed for each study using the QUADAS-2 tool, which is given in supplemental Appendix B and Appendix C. Notably, there was a high risk of bias related to patient selection in 66% (10/15) of studies, while in the remaining five, the risk was unclear. Regarding concerns of study applicability, these were high or unclear in 53% (8/15) of studies for patient selection. Six studies (5/15, 33%) did not explicitly mention their inclusion or exclusion criteria, and 3/15 (20%) studies excluded patients based on factors that could increase selection bias, such as IA size, type, location, or presence of comorbidities.

### 4.4. Current State of the Art Performance

Among the 15 studies analyzed, 6 studies (40%) demonstrated a seemingly favorable trade-off between lesion-level sensitivity (greater than 0.8) and the number of false positives per image (less than 2) [9,12,19,21,22,26]. According to the FROC in Figure 4, Wang et al. [19] achieved the best trade-off between the lesion-level sensitivity and false positives per image, with respective values being 0.944 and 0.6. However, the authors did not explicitly provide information on the average size of the utilized IAs (in Figure 4, we used a weighted estimate based on IA size categorization and associated counts as weights), and their test dataset included only 2 aneurysms smaller than 3 mm, none of which were detected by their method.

### 4.5. Size-Based Lesion-Level Sensitivity

A positive trend has been observed in reporting sensitivity based on the size of IAs. Eight (8/15, 53%) studies reported sensitivity based on the IA size categorization, and two studies reported sensitivity based on volume [9,25]. Although volume-based reporting may offer greater accuracy, it is not widely reported, and the IA volume is not used in clinical guidelines for IAs, likely due to the need to use advanced tools to enable its assessment.

Although there is a consensus among different authors regarding the threshold for small IAs (diameter < 3 mm), there is no agreement for medium and large IA. Among the studies that reported IA size categorization, 5 out of 8 (5/8, 62%) studies used three categories, while the other 3 studies (3/8, 38%) defined four categories. All definitions with associated size thresholds can be found in Figure 3.

The most commonly used categories were <3 mm for small, 3–7 mm for medium, and >7 mm for large IAs, and we recommend reporting sensitivity for each of these three categories in any future studies. By adopting these proposed group sizes, future studies can achieve a higher level of consistency and comparability when reporting lesion-level sensitivity based on IA size.

In the detection of small IAs, the highest lesion-level sensitivity (0.926) was reported by Yang et al. [8]. However, their method also exhibited the highest FPs/image (13.8) among all studies. In contrast, Wei et al. [23] achieved a low FPs/image of 0.165 while attaining overall lesion-level sensitivity of 0.77, but a lower lesion-level sensitivity of 0.548 for small IAs. This finding further underscores the inherent trade-off between the number of FPs/image and sensitivity, particularly in the context of detecting small IAs. Furthermore, all mentioned results may be over-optimistic due to the use of internal test datasets, as opposed to external ones that generally involve site, scanner, or imaging protocol bias that adversely affects the detection performance.

### 4.6. Guidelines on Evaluation Metrics

To ensure consistent reporting of the performance in future studies involving IA detection, we have compiled a set of required metrics:

**Lesion-Level Sensitivity**: This metric measures the number of correctly detected IAs divided by the total number of aneurysms in the dataset. The authors should report the overall lesion-level sensitivity and further split it into three groups: small (<3 mm), medium (>3 mm and <7 mm), and large (>7 mm). Such split is recommended for consistent reporting purposes (refer to Figure 3). Furthermore, the methods and tools of IA size measurement should be described, along with the qualification and years of experience of the rater.

**Patient-Level Sensitivity**: This metric evaluates the number of correctly segmented scans, where all IAs on the scan are detected, divided by the total number of scans that include IAs. When the model fails to detect one or more IAs in a scan, such scan is labeled as a false negative. Conversely, if the model successfully detects all IAs in the scan, such scan is labeled as a true positive.

**Patient-Level Specificity**: This metric assesses the number of correctly labeled scans from healthy subjects divided by the total number of scans that do not depict IAs. Each study needs to include healthy subjects to evaluate the model’s ability to accurately identify cases without IAs. Correctly labeled scans of healthy subjects are defined as scans without any false positive findings.

**Number of False Positives (FPs) per Image**: This metric quantifies the number of false positive IAs detected in the entire dataset, divided by the total number of scans depicting the IAs. It provides valuable information regarding the rate of false positive detections. The same metric can be assessed and reported on healthy subject scans and reported in conjunction with patient-level specificity.

By consistently reporting these metrics, researchers can facilitate comparisons between different studies and, therefore, enhance the understanding of IA detection performance.

## 5. Conclusions

In summary, our analysis highlights the need for standardized reporting guidelines in studies involving the detection of IA using Computed Tomography Angiography (CTA). The existing literature demonstrates promising results in IA detection using CTA, but limitations such as high risk of bias, inconsistent reporting of evaluation metrics, and lack of healthy subject control group need to be addressed. Moving forward, it is crucial to establish uniform reporting guidelines to facilitate meaningful comparisons among future studies. We provided guidelines on metrics for IA detection, including lesion-level sensitivity, patient-level sensitivity, patient-level specificity, and the number of false positives per case. By consistently reporting these metrics, researchers can improve the reproducibility and comparability of IA detection studies.

The use of a common evaluation dataset is a further step toward objective and comparative evaluation of IA detection performance. We highly recommended that all authors employ the dataset provided by Bo et al. [24] (and any other future public datasets) as an external validation dataset to facilitate comparisons between different IA detection methods.

Additionally, our findings underscore the importance of including healthy subjects in IA detection studies to mitigate spectrum bias and enhance the generalizability of the results. Specifically, standardized reporting of performance should include per-subject specificity on the control group to highlight the rate of false findings. To the best of our knowledge, a public dataset of healthy subject CTAs is not yet available, and its collection remains in the domain of the authors of future studies.

Overall, standardizing reporting guidelines and addressing the limitations identified in this analysis will contribute to the advancement of IA detection on CTA, and possibly on other modalities, and provide more robust evidence for possible adoption of IA detection methods in computer-assisted diagnosis, triage, and decision-making systems.

## Figures and Tables

**Figure 1 biomedicines-11-02921-f001:**
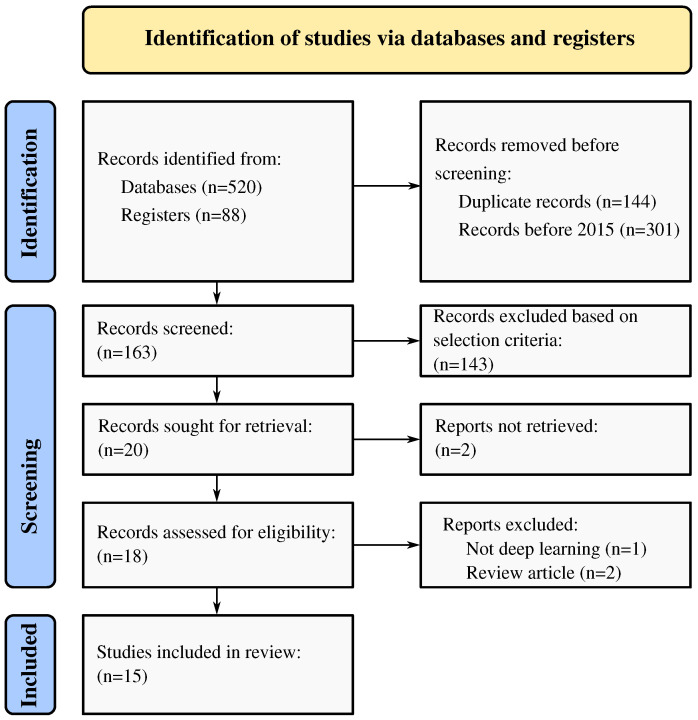
Flow diagram for systematic review and meta-analysis of cerebral aneurysm detection using artificial intelligence.

**Figure 2 biomedicines-11-02921-f002:**
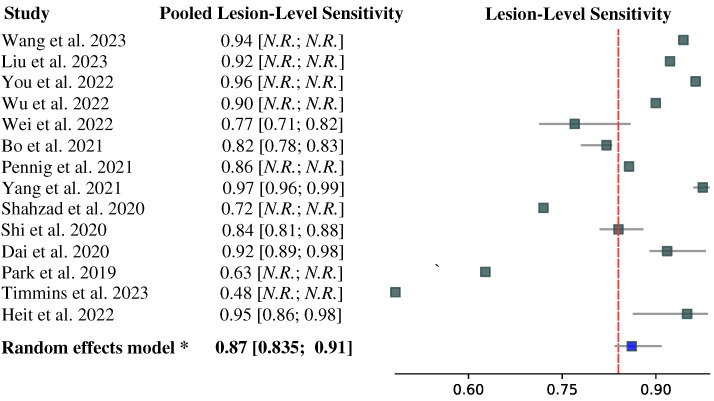
Forest plot of pooled lesion-level IA detection sensitivity from papers [8,9,19,20,21,22,23,24,25,26,27,28,29,30]. The 95% confidence intervals (CIs) are given in *square brackets*. If CIs were not given in the original study, we reported herein the point estimates of the pooled lesion-level sensitivity.

**Figure 3 biomedicines-11-02921-f003:**
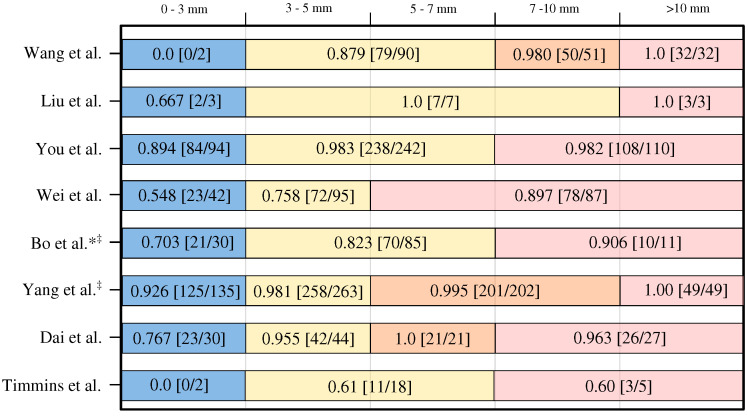
Lesion-level sensitivity with respect to the IA size category (noted on top) [8,17,20,21,23,24,27,29]. * means no exact number of cases per size reported; approximations derived from Figure 3 in Bo et al. paper [24]. ^‡^ means authors used external validation but reported size-based performance on internal dataset.

**Figure 4 biomedicines-11-02921-f004:**
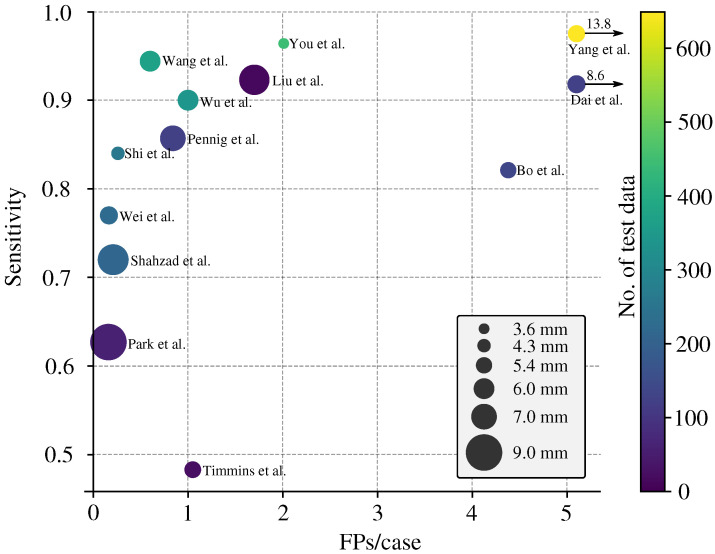
Enhanced FROC with lesion-level sensitivity on *the vertical axis* versus the number of False Positives (FPs) per image on the *horizontal axis*. Each data point represents the result from one study [8,9,19,20,21,22,23,24,25,26,27,28,29], whereas its color represents the number of test images, and size indicates the average IA size.

**Table 1 biomedicines-11-02921-t001:** Summary of included studies. Most studies used automated detection approaches, except those in *bottom two rows*, which used a semi-automated approach.

Study	Publication Year	Modalities	Data Source	Deep Learning Model	Control Group	Data Availability
Wang et al. [19]	2023	CTA	multicenter	DAResUNet	yes	no
Liu et al. [20]	2023	CTA	multicenter	Deep Medic	no	U.R.
You et al. [21]	2022	CTA	multicenter	3D-UNet	no	U.R.
Wu et al. [22]	2022	CTA	multicenter	Dual-channel ResNet	no	U.R.
Wei et al. [23]	2022	CTA	single center	ResUNet	no	no
Bo et al. [24]	2021	CTA	multicenter	GLIA-NET	yes	yes
Pennig et al. [9]	2021	CTA	single center	Deep Medic	no	U.R.
Meng et al. [31]	2021	CTA	single center	N.R.	no	no
Yang et al. [8]	2021	CTA	multicenter	3D DLN-OR	yes	no
Shahzad et al. [25]	2020	CTA	single center	Deep Medic	no	no
Shi et al. [26]	2020	DSA, CTA	multicenter	DAResUNet	yes	no
Dai et al. [27]	2020	CTA	single center	ResNet	no	no
Park et al. [28]	2019	CTA	single center	HeadXNet	yes	no
Timmins et al. [29]	2023	CTA, MRA	single center	MESH CNN	no	no
Heit et al. [30]	2022	CTA	multicenter	N.R.	no	no

N.R. = Not Reported; U.R. = Upon request.

**Table 2 biomedicines-11-02921-t002:** Detection performance summary, using data extracted from each study. Most studies used automated detection approaches, except those in *bottom two rows*, which used a semi-automated approach.

Study	No. of CTA Scans	No. of IA (Train/Test)	Lesion-Level Sensitivity	Patient-Level Specificity	FPs per Image	Average Size	Size Split
Wang et al. [19]	1547	2037 (1667/175 + 195 ^★^)	0.944	N.R.	0.6	N.R.	yes
Liu et al. [20]	90	112 (98/13)	0.923	N.R.	1.7	7.9	yes
You et al. [21]	2272	2938 (2492/446)	0.964	N.R.	2.01	3.6	yes
Wu et al. [22]	1508	1710 (1370/340)	0.900	N.R.	1	6.0	no
Wei et al. [23]	212	224 (/224)	0.77	N.R.	0.165	5.4	yes
Bo et al. [24]	1476	1590 (1363/126 + 101 ^★^)	0.821	N.R.	4.38	5.0	yes
Pennig et al. [9]	172	205 (79/126)	0.857	N.R.	0.84	R.V.	no
Meng et al. [31]	100	N.R.	N.R.	N.R.	N.R.	N.R.	no
Yang et al. [8]	1068	1543 (688/649 + 206 ^★^)	0.975	N.R.	13.8	5.2	yes
Shahzad et al. [25]	253	294 (79/215)	0.72	N.R.	0.21	R.V.	no
Shi et al. [26]	1313	1676 (1099/314 + 263 ^★^)	0.84	0.71	0.26	4.3	no
Dai et al. [27]	311	344 (222/122)	0.918	N.R.	8.6	5.4	yes
Park et al. [28]	818	328 (269/59)	0.627	0.06	0.16	no	no
Timmins et al. [29]	20	25 (^†^/25)	0.483	N.R.	1.05	5.1	yes
Heit et al. [30]	51	60 (0/60)	0.95	N.R.	N.R.	5.4	no

N.R. = Not Reported; R.V. = Reporting Volume; ${}^{★}$ = external dataset; ${}^{†}$ = Trained on MRA.

## Data Availability

Not applicable.

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
