# Peer review of "A Systematic Review of Deep-Learning Methods for Intracranial Aneurysm Detection in CT Angiography"

_biomedicines, 2023, doi:10.3390/biomedicines11112921_

Round 1

Reviewer 1 Report

Comments and Suggestions for Authors

A good and actual systematic review. Questions were selected adequately, data selection, analysis and conclusions logical. Illustrations good. Although one methodological questions is not included - what deep learning algorithms were utilized in the studies? This factor can have some influence to variability of the results?

Author Response

We thank you for a comprehensive feedback on our submission and constructive comments, which helped us improve the contributions, presentation and clarity of the manuscript. We have carefully revised the original manuscript by addressing all of the concerns raised.

Comment #R1.1:

Although one methodological questions is not included - what deep learning algorithms were utilized in the studies? This factor can have some influence to variability of the results?

Response:

We appreciate your valuable feedback. We indeed gathered information about the deep learning models employed in the reviewed studies. The utilized deep learning algorithms are listed in Table 1. Furthermore we included following paragraph in section Results:

In the examined studies, 5 (5/15, 33\%) studies made use of either the original ResNet model or a modified variant thereof, and 3 (3/15, 20\%) adopted the Deep Medic model. Meanwhile, various other models such as 3D-UNet, GLIA-NET, 3D DLN-OR, and HeadXNet were each employed only once. Notably, two authors did not specify the deep learning model they used.

Moreover, we've taken your suggestion into account and changed a paragraph in Discussion from:

One significant limitation of all the reviewed studies is that each study validated their DLM exclusively on their private datasets. This is mainly because, until recently, there were no publicly available CTA datasets of IAs. Since the authors may not release their code and datasets for various reasons, for instance due to patient privacy policies and other regulatory limitations, it is therefore impossible to independently verify the effectiveness of their methods.

to:

One significant limitation of all the reviewed studies is that each study validated their DLM exclusively on their private datasets. This is mainly because, until recently, there were no publicly available CTA datasets of IAs. Since the authors may not release their code and datasets for various reasons, for instance due to patient privacy policies and other regulatory limitations, it is therefore impossible to independently verify the effectiveness of their methods and assess the variability of results with respect to the employed deep learning model architecture."

Kind regards,

Žiga Bizjak and Žiga Špiclin

Reviewer 2 Report

Comments and Suggestions for Authors

This manuscript represents a report detailing a systematic review and meta-analysis of studies related to the detection of intracranial aneurysms through the utilization of artificial intelligence. Notably, the manuscript contains insights pertaining to potential areas of enhancement and provides guidance for the future direction of research endeavors.

 I have a minor comment on the abstract.

Indeed, it appears reasonable to assert that the progression of artificial intelligence systems is likely to improve the detection rate of unruptured intracranial aneurysms. However, there persists uncertainty (disagreement) regarding whether this will lead to better patient outcomes, as the quality of diagnosis differs from that of treatment.

Author Response

We thank you for a comprehensive feedback on our submission and constructive comments, which helped us improve the contributions, presentation and clarity of the manuscript. We have carefully revised the original manuscript by addressing all of the concerns raised.

Comment #R2.1:

Indeed, it appears reasonable to assert that the progression of artificial intelligence systems is likely to improve the detection rate of unruptured intracranial aneurysms. However, there persists uncertainty (disagreement) regarding whether this will lead to better patient outcomes, as the quality of diagnosis differs from that of treatment.

Response:

Indeed, while it is reasonable to anticipate that the advancement of artificial intelligence systems could potentially enhance the detection rate of unruptured intracranial aneurysms, it is crucial to recognize the distinction between diagnosis and treatment quality. The impact on patient outcomes involves multifaceted factors beyond just improved detection accuracy. We have taken you suggestion into account and revised the related abstract sentence. The original sentence:

"Early identification of aneurysms on Computed Tomography Angiography (CTA), a frequently used modality for this purpose, is crucial, and artificial intelligence (AI) based algorithms can significantly improve patient outcomes by aiding in their detection."

now reads as:

"Early identification of aneurysms on Computed Tomography Angiography (CTA), a frequently used modality for this purpose, is crucial, and artificial intelligence (AI) based algorithms can improve the detection rate and minimize the intra- and inter-rater variability.”

Kind regards,

Žiga Bizjak and Žiga Špiclin

Reviewer 3 Report

Comments and Suggestions for Authors

The authors presented a well written and illustrated article performing a systematic review on the use of AI and deep learning method to detect aneurysms on CTA. 

Although the article is clear and methodology is sound, it substantially repeat very recent reviews that have been recently published on the matter, and that have been also cited by the authors in the introduction. It is not clear how a systematic review concentrated only on CTA (while the previous included also MRA and DSA) could give significant more data compared to the existing literature. Furthermore, from a clinical point of view, MRA is usually the most used methodology to screen patients for the presence of intracranial aneurysms, because it is considered less invasive (absence of radiation and no contrast needed). Therefore, from a clinical point of view, it would have been much more interesting to concentrate their effort on MRA rather than CTA only. 

Author Response

We thank you for a comprehensive feedback on our submission and constructive comments, which helped us improve the contributions, presentation and clarity of the manuscript. We have carefully revised the original manuscript by addressing all of the concerns raised.

Comment #R3.1:

Although the article is clear and methodology is sound, it substantially repeat very recent reviews that have been recently published on the matter, and that have been also cited by the authors in the introduction. It is not clear how a systematic review concentrated only on CTA (while the previous included also MRA and DSA) could give significant more data compared to the existing literature.

Furthermore, from a clinical point of view, MRA is usually the most used methodology to screen patients for the presence of intracranial aneurysms, because it is considered less invasive (absence of radiation and no contrast needed). Therefore, from a clinical point of view, it would have been much more interesting to concentrate their effort on MRA rather than CTA only.

Response:

In our manuscript, we have cited two prior meta-reviews of aneurysm detection, one by Gu et al. and the other by Din et al. Both meta-review papers encompassed aneurysm detection methods across the three modalities: MRA, DSA, and CTA. Notably, during our assessment of the Din et al.'s meta-review, we observed the inclusion of 18 MRA, 8 DSA, 2 multimodality, and only 3 CTA methods, spanning the years from 2004 to 2021. The meta-review by Gu et al., on the other hand, featured 6 MRA, 5 DSA, and 6 CTA methods. While they encompassed a significant portion of the available literature up to that point, the reporting of detection metrics, especially for the CTA modality, was rather limited. In our study, we extended the scope by examining a more substantial number of methods, reviewing a total of 15 papers, indicating recent rapid progress in the field. We acknowledge that the rationale behind our manuscript could be better elucidated. To address this, we have rephrased a paragraph about the rationale in the Introduction. In the original version of the manuscript we wrote:

"Recent two review studies focused on all three commonly used modalities (MRA, CTA and DSA) for IA detection, but have not provided a comprehensive review and analysis of methods utilizing the CTA modality in their test datasets. To address this gap and enhance the understanding of IA detection using CTA, we have performed a systematic review and meta-analysis, using established methodologies. This study seeks to contribute to the existing knowledge and provide a comprehensive review of the performances of DLMs in the detection and diagnosis of IAs using CTA, specifically focusing on test dataset structure and size and consistency and validity of the reported evaluation metrics. This will elucidate the current state of developments in the field, offering crucial guidance for shaping future research endeavors and, ultimately, influencing evidence-based clinical practices."

In the revised manuscript, the particular paragraphs now reads as:

"Recent two review studies focused on all three commonly used modalities (MRA, CTA, and DSA) for IA detection, but have not provided a comprehensive review and analysis of methods utilizing the CTA modality in corresponding test datasets. Specifically, Din et al. reviewed 3 and Gu et al. reviewed 6 research papers, while our review of the field identified 15 eligible research papers, thus indicating recent rapid progress in the field since the aforementioned two studies were published. To enhance the understanding of state of the art in IA detection using CTA, we have therefore performed a systematic review and meta-analysis, using established methodologies. We specifically focus on the structure and size of test datasets employed in the reviewed studies, as well as the consistency and validity of the reported evaluation metrics. By examining a larger pool of studies, our analysis aims to elucidate the current state of developments in the field, offering crucial guidance for shaping future research endeavors and, ultimately, influencing evidence-based clinical practices.”

Furthermore, we concur with the Reviewer's viewpoint that MRA is frequently the methodology of choice for screening patients for intracranial aneurysms. However, it's worth noting that due to the efficiency and affordability of CTA, it continues to be extensively employed for cerebrovascular imaging

Kind regards,

The authors